# Position: Let's Develop *Data Probes* to Fundamentally Understand How Data Affects LLM Performance

Shiqiang Wang [1]   Herbert Woisetschläger [2]   Hans Arno Jacobsen [3]   Mingyue Ji [4]

## Abstract

Data is fundamental to large language models (LLMs). However, understanding of what makes certain data useful for different stages of an LLM workflow, including training, tuning, alignment, in-context learning, etc., and why, remains an open question. Current approaches rely heavily on extensive experimentation with large public datasets to obtain empirical heuristics for data filtering and dataset construction. These approaches are compute intensive and lack a principled way of understanding the essence of how specific data characteristics drive LLM behavior. In this position paper, *we advocate for the need of developing systematic methodologies for generating synthetic sequences from appropriately defined random processes, with the goal that these sequences can reveal useful characteristics when they are used in one or multiple stages of the LLM workflow*. We refer to such sequences as *data probes*. By observing LLM behavior on data probes, researchers can systematically conduct studies on how data characteristics influence model performance, generalization, and robustness. The probing sequences exhibit statistical properties that can be viewed using theoretical concepts, such as typical sets, which are generalized to describe the behaviors of LLMs. This data-probe approach provides a pathway for uncovering foundational insights into the role of data in LLM training and inference, beyond empirical heuristics.

## 1. Introduction

The success of large language models (LLMs) stems not only from advances in model architecture and training algorithms but also from access to vast amounts of diverse data (Grattafiori et al., 2024; Mishra et al., 2024; Brown et al., 2020; DeepSeek-AI, 2025). Raw data typically cannot be used directly. It must be processed or filtered to ensure quality. Such data processing is resource-intensive (Penedo et al., 2024), so, as of today, data-related research has been primarily conducted by large organizations with the necessary computational and financial resources (Grattafiori et al., 2024; Gemma Team et al., 2024; Su et al., 2025; Gohari et al., 2026). While such research has been practically useful, there is limited study on the *fundamental reasons behind how data affects LLM performance*.

Beyond the need for extensive resources, another obstacle preventing the understanding of data for LLMs is the lack of a systematic and principled way of controlling the data input for LLM training and inference. Real-world data is often not directly controllable, as its ground-truth distribution is largely unknown (Cvejoski et al., 2024; Shu & Yu, 2024). Thus, current data processing methods rely on empirical heuristics, which are developed through extensive experimentation involving LLM training with data processed in different ways and evaluation on benchmark datasets (Wettig et al., 2024; Penedo et al., 2024). Such empirical findings usually hold only on a case-by-case basis. Moreover, the training data may be contaminated with benchmark data, and the benchmark datasets themselves may not fit the target domain where the LLM will be applied (Sainz et al., 2023).

Therefore, an important *open problem* is the fundamental relationship between data properties and LLM behavior. Addressing this problem is crucial, as a better understanding of how data affects LLM performance could lead to more efficient and targeted dataset construction, reducing costs and risks (e.g., hallucination) while improving the overall LLM performance. Recently, some theoretical studies have attempted to use simplified sequences to analyze specific properties of transformer-based architectures (Makkuva et al., 2025; Rajaraman et al., 2024). Although these studies provide valuable insights, their evaluations usually consider oversimplified models and have limited relevance to practical LLM workflows. What is missing is a *systematic yet ac-*

---
[1]Department of Computer Science, University of Exeter, UK [2]Technical University of Munich, Germany [3]Department of Electrical and Computer Engineering, University of Toronto, Canada [4]Department of Electrical and Computer Engineering, University of Florida, FL, USA. Correspondence to: Shiqiang Wang <s.wang9@exeter.ac.uk>, Herbert Woisetschläger <h.woisetschlaeger@tum.de>.

*Proceedings of the 43ʳᵈ International Conference on Machine Learning*, Seoul, South Korea. PMLR 306, 2026. Copyright 2026 by the author(s).

*cessible approach that can connect theoretical findings with real-world applications*, offering both explanatory power and actionable guidance for practitioners.

In this paper, we argue that **the research community should develop systematic methodologies for generating synthetic sequences from fully defined (known) random processes, with the goal that these sequences will be useful tools for understanding the relationship between data and LLM performance. We refer to such sequences as** *data probes*. The data probes are designed to have a clear meaning (usually in theory) while being able to trigger specific behaviors in practical LLMs. Different data probes can be used depending on the LLM behavior, e.g., hallucination, bias, memorization, mode collapse, we would like to study.

Compared to actual datasets, the uniqueness of data probes is that they are generated from a random process with a *known* probability distribution. Although it may be high-dimensional and difficult to fully comprehend from a human perspective, the fact that this distribution is known and can be expressed numerically has the following advantages:

1. A virtually unlimited amount of data can be generated from the same distribution, so both training and test data can be generated and used in experiments driven by data probes. Such data can be generated on the fly, removing the need to manage massive datasets.

2. The likelihood of any given sequence can be computed against the distribution, which is impossible for experiments with real datasets where the underlying stochastic process for generating such real data is unknown. This unlocks new opportunities, such as examining the difference in the likelihood between the training data and LLM-generated data, which subsequently help advance the research in LLMs and generative AI.

In essence, by systematically varying key characteristics of the probability distribution for generating data probes, researchers can observe how these properties influence LLM performance, where the results can be obtained by evaluating LLM-generated data back on the known distribution. Compared to standard empirical studies driven by massive datasets, this approach is more controllable and requires significantly fewer resources. It can reveal the *core principles* underlying the influence of data on LLMs. The results of such data probe-based studies will be *a valuable starting point for more sophisticated theoretical analysis and practical design of data processing algorithms*.

Data probes also allow for deeper integration of theoretical concepts, such as typical sets from information theory (Cover & Thomas, 2006), thus offering a principled framework. However, the difference between this data-probe approach and most existing theoretical analyses is that *data probes are designed to have both theoretical and practical values*. The results provide actionable insights into how datasets can be better constructed and curated. In this way,

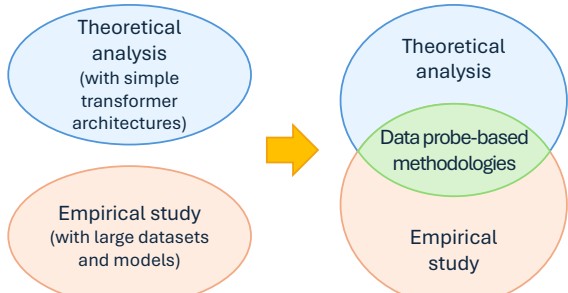

*Figure 1.* Data probes connect theory and practice.

data probes will be *an important "interface" for connecting theory and practice*, as illustrated in Figure 1.

This position paper highlights the importance of developing and adopting data probes to advance both research and practice. They offer a powerful way to bridge the gap between theory and real-world application, helping the community develop more efficient, transparent, and effective strategies for leveraging data in developing next-generation LLMs. This approach can also inspire a closer collaboration and integration of data-related research between academic institutions and large industrial organizations.

## 2. Current Status of Understanding LLMs

The understanding of LLMs has evolved through various complementary approaches, each offering unique insights into these complex systems. While these approaches have advanced our knowledge significantly, they also reveal important gaps that the data-probe methodology can address.

**Datasets and Benchmark Tasks.** The development and evaluation of LLMs has been largely driven by performance on standardized benchmarks. These benchmarks span diverse tasks including question answering, natural language inference, and text generation (EleutherAI, 2026; Chiang et al., 2024; Zheng et al., 2023). However, benchmark-based evaluation often do not explain *why* models succeed or fail for specific types of samples.

**Physics of LLMs.** Recent works have explored using physics-inspired principles to understand the behavior and capabilities of LLMs, by dissecting their abilities across different domains such as hierarchical structure understanding, reasoning, factual knowledge management, and scaling laws (Allen-Zhu, 2024; Kaplan et al., 2020; Hoffmann et al., 2022; Wang et al., 2025). While some works use specifically hand-crafted training data to study specific problems in an experimental setting, which to some degree follows a similar spirit as data probes, they generally do not leverage the statistical distribution of the data. Some data creation mechanisms used in physics of LLMs may be overly specific to the problems being studied, making them difficult to be reused for other problems or conduct theoretical analyses.

**Mechanistic Interpretability.** Mechanistic interpretability aims to understand LLMs by reverse-engineering their inter-

nal representations and decision-making processes (Singh et al., 2024; Räuker et al., 2023). This includes techniques for analyzing attention patterns, identifying specific neurons or circuits responsible for particular behaviors, and tracking the flow of information through model layers. Recent advances have revealed specific mechanisms for tasks like induction heads and mathematical operations, which provide concrete insights into how LLMs process and manipulate information, but there is limited understanding on how different types of training data lead to the development of these mechanisms.

**Theoretical Analysis of Transformer Models.** Recent studies have explored the theoretical aspects of transformer models, focusing on their ability to handle tasks like learning patterns, capturing long-range dependencies, and processing hierarchical structures (Edelman et al., 2024; Makkuva et al., 2025; Rajaraman et al., 2024; Von Oswald et al., 2023; Zekri et al., 2024). These analyses often use simplified transformer architectures to make the theory more manageable. Although they help explain how transformers work in theory, their connection to real-world LLMs is limited, and some important aspects such as how specific data characteristics influence model behavior may be overlooked.

## 3. Why Data Probes Need to Be Developed

Data probes offer a way to systematically vary distributional properties in a controlled environment while still connecting to real-world LLMs. They are important because they introduce a structured and controllable way to study data itself, treating it as a formal object with known statistical properties rather than as a fixed input. By generating sequences from explicitly defined random processes, data probes enable the precise specification of distributional characteristics such as entropy, mutual information, or temporal correlation. This level of control is not achievable with natural language data, where the true underlying distribution is unknown and difficult to model. Moreover, as data probes are drawn from known generative mechanisms, they allow for reproducible experimentation, tractable analysis, and direct integration with formal frameworks such as information theory. With these characteristics, data probes can address the following limitations in current approaches.

**Enabling Resource-Efficient Experiments.** Much of the difficulty in studying how data affects LLMs stems from the immense cost of assembling and managing massive real-world datasets (Lozhkov et al., 2024; Penedo et al., 2024; Weber et al., 2024). Data probes circumvent these costs by generating synthetic sequences on the fly, which can be scaled to arbitrary sizes without additional storage or curation efforts. Researchers and practitioners can thus rapidly iterate over numerous distributions, testing hypotheses about data quality and diversity with less computational resources.

**Connecting Theory and Practice.** Data probes serve as a "bridge" between theoretical analyses and practical LLMs by allowing researchers to design controlled distributions that can be scaled in complexity. In this way, one can evaluate whether theoretical predictions hold in practice and iteratively refine both theory and data-probe design.

**Understanding the Effect of Data Characteristics.** While benchmarks highlight *what* models can do, they rarely explain *why* they succeed/fail. Data probes make it possible to isolate statistical properties and analyze how they influence learning and generalization. As the underlying distribution is known, one can compute statistics, e.g., likelihood, of generated sequences on the true data distribution, revealing insights into memorization or under-represented patterns.

**Reducing Noise in Empirical Studies.** Real-world datasets have properties like domain imbalance and annotation artifacts, making it difficult to disentangle genuine model capabilities from spurious cues in the data (Gardner et al., 2021; Gururangan et al., 2018). In contrast, data probes can be designed to eliminate or control these limiting factors, allowing for cleaner experiments that pinpoint the roles of different data features.

**Extending Beyond Benchmark Coverage.** Although benchmarks have fueled progress in LLM research, they naturally lag behind the breadth and complexity of real-world language tasks (Chollet et al., 2024; Fourrier et al., 2024; Wang et al., 2019). Data probes can complement benchmark suites by creating new challenge sets that target under-explored aspects of language complexity, e.g., rare syntactic patterns and unusual compositional structures. Such targeted testing can reveal performance bottlenecks or emergent capabilities that standard benchmarks might miss.

**Guiding Data Processing and Model Development.** Current methods for evaluating LLM performance beyond standard benchmarks rely on either expensive gold-standard datasets (Chelli et al., 2024; Pacchiardi et al., 2024) or approaches that trade off speed and cost for quality by using weighted outputs from other LLMs (Sam et al., 2025; Huang et al., 2025; Wei et al., 2024). Both approaches are reactive, i.e., they assess performance after training is complete. Data probes offer a fundamentally different approach by identifying how specific distributional features in training data correlate with downstream performance and interpretability. This enables proactive principled guidance for dataset selection and filtering before training, while also informing critical decisions about model architecture and training curricula. It is a path to more robust and efficient LLMs through data-driven optimization rather than post-hoc evaluation.

Overall, data probes bring a new level of control and clarity to investigating LLM behavior, reducing the reliance on trial-and-error approaches with large heterogeneous datasets. They complement existing studies (summarized in Section 2) by providing a unified framework that is theoretically grounded yet applicable to large-scale real-world models.

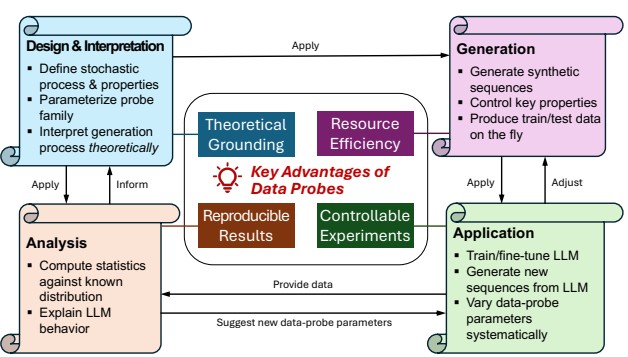

*Figure 2.* Overview of the data-probe methodology.

# 4. How to Design and Use Data Probes

## 4.1. Overall Methodology

Figure 2 provides an overview of the methodology. The first step of designing data probes involves defining a generative process with theoretical interpretation, along with controllable statistical properties that can be systematically varied and measured. Effective data-probe design is an open research area, which can include characterizing relationships between probe structure and model capacity, developing informativeness metrics, and identifying minimal configurations that expose specific behaviors. Over time, this could yield a collection of probe families optimized for different investigations. Once the data-probe generation process is designed, the probes are generated and applied to LLM training or fine-tuning processes. The data generated from the probe-trained LLM is then used for analysis. These steps are coupled with each other, e.g., the choice of probes can vary depending on the observations made throughout the experimentation. The next subsection provides an example of how these steps can coherently work together.

## 4.2. Operational Definition and Validation Protocol

We define a data probe as

$$\Pi = (\mathcal{P}, \mathcal{M}, \mathcal{H}, \mathcal{F}), \tag{1}$$

where $\mathcal{P}$ specifies a known generative process and intervention controls, $\mathcal{M}$ is a set of measurable diagnostics, $\mathcal{H}$ is a set of testable claims, and $\mathcal{F}$ specifies falsification rules.

We define four core criteria with validity predicates:

- $C_1$: Known process is fully specified and samplable.
- $C_2$: Controllable knobs and interventions are defined and interpretable.
- $C_3$: Diagnostic metrics are computable.
- $C_4$: Each claim has pre-declared falsification conditions.

Evaluation uses two layers. Internal validity (denoted by $IV(h)$) checks whether probe-side directional predictions hold under interventions. External validity (denoted by $EV(h)$) checks whether matched real-side directional effects and falsification tests hold. For a claim $h \in \mathcal{H}$, transfer is accepted if and only if

$$\text{Accept}(h) = 1 \iff IV(h) = 1 \wedge EV(h) = 1. \tag{2}$$

---

**Claim Validation and Transfer Decision Logic**
**Prerequisites:** satisfy $C_1$–$C_4$ and document reduction assumptions.
**Internal validity check:** evaluate $IV(h)$ in probe space under declared intervention contrasts.
**External validity check:** evaluate $EV(h)$ in matched real settings under the same claim direction.
**Decision matrix:**
$IV(h) = 1$, $EV(h) = 1 \Rightarrow$ transfer supported
$IV(h) = 1$, $EV(h) = 0 \Rightarrow$ probe-local result
$IV(h) = 0 \Rightarrow$ claim rejected under declared criterion

*Figure 3.* Validity and transfer decision logic for a claim $h$.

If $IV(h) = 1$ but $EV(h) = 0$, the result is probe-local. This pass/fail structure makes transfer claims falsifiable rather than narrative. A formal object, predicate definitions, and transfer equations are provided in Appendix B. Figure 3 illustrates the logic.

**Two Entry Paths under One Protocol.** Bottom-up starts from a theoretical mechanism and builds probes to test whether predicted effects appear in practice. Top-down starts from a practical failure pattern and reduces it into a tractable probe while documenting removed factors and preserved properties. The two paths differ in entry point, but both use the same claim specification, documentation step, and IV/EV transfer decision rule.

## 4.3. Specific Example

In the following, we provide a *basic example* illustrating how data probes can be generated and used for revealing characteristics of LLMs. *More advanced data probe designs and experiments are beyond the scope of this position paper and are worth studying in future work.*

We consider a pair of models with the GPT-2 small transformer architecture (Radford et al., 2019b). We use a Markov chain[1] with a given *entropy rate* to generate data probes that serve as training data for the first model, hereafter referred to as probe-LLM, where the embedding layer of this model is tailored to the vocabulary size (state space) of the Markov chain. The second model, referred to as text-LLM, is the pre-trained `openai-community/gpt2` model from Huggingface (Radford et al., 2019a). The entropy rate, expressed in *bits per token*, specifies the diversity of the generated data probes, where higher entropy rates produce more varied sequences, and lower entropy rates produce more predictable sequences. We also use the same Markov chain to measure the likelihood of sequences generated by probe-LLM after training, demonstrating how the known distribution can be leveraged in our experiments. To show the effectiveness of data probes in understanding fundamental dynamics with real models, we connect the insights observed on probe-LLM and text-LLM. More details about this experiment are provided in Appendix C.

Our exemplar procedure mainly consists of these parts:

---

[1]Note that data probes are *not* limited to Markov chain-generated sequences. We consider a Markov chain in this example for its simplicity and sufficiency for illustration purposes.

*Table 1.* Reduction record for the current basic example. Each row records a reduction operator, preserved invariants, expected directional hypothesis, and protocol-level rejection condition, together with execution status in this basic study.

| Step | Reduction operator | Preserved invariants | Directional hypothesis | Rejection condition | Status |
|------|--------------------|----------------------|------------------------|---------------------|--------|
| 1 | Map open-domain corpus to a Markov probe generator | Entropy rate and transition law are defined and controllable | Generated outputs admit stable typical-set regime assignment | Regime assignment does not separate decoding conditions in probe space | Checked |
| 2 | Remove semantic and topic factors from training distribution | Sequence-likelihood structure remains defined by known law | Average NLL is a calibrated diagnostic linked to the generator | Average NLL does not track likelihood under the known Markov law | Checked |
| 3 | Fix architecture & checkpoint and intervene only on temperature $T$ | Single-knob intervention isolates decoding diversity effects | As $T$ increases, regime mass shifts from over-conservative to typical then to uncertain | Probe-side regime-shift direction is not observed under the declared contrast | Checked |
| 4 | Use text-LLM outputs as matched real-side comparator | Intervention direction is shared across probe and real settings | Real-side degeneration and diversity trend aligns directionally with probe-side trend | No matched directional trend in real-side outputs under the same intervention ordering | Qualitative only |

1. We **document** reduction decisions first, using a reduction record that lists removed factors, preserved statistical properties, expected effects, and falsification conditions.
2. We **design** a probe-generation mechanism based on a Markov chain with a desired entropy rate, and **generate** token sequences from this Markov chain. The sequences are then used as data probes.
3. We **theoretically interpret** this design with the concept of typical sets in information theory, providing a theoretical lens for examining the likelihood of token sequences.
4. We **apply** data probes to train probe-LLM, then **analyze** probe-LLM's generation behavior and compare it with that of text-LLM, using both greedy decoding and temperature-based sampling.

The reduction steps are in Table 1. This record makes it clear how the data-probe setup is derived from a more realistic setting and what assumptions are required for interpretation. In this basic example, probe-side conditions are checked, while the real-side comparator is qualitative, so some rejection conditions remain protocol-level targets for a full transfer study.

**Design Based on Markov Chain with Target Entropy Rate.** Since it is generally difficult to directly generate a Markov chain with a given entropy rate, denoted by $H$, we design our probe generation process so that it first generates random transition matrices for a Markov chain and then select the one that gives an entropy rate closest to $H$. The details are provided in Appendix D.

**Theoretical Interpretation via Typical Sets.** As part of studying LLMs with data probes, we can connect the data-probe generation process and its underlying distribution to a theoretical framework for better interpretation and in-depth study. In this example, we make a connection with the notion of typical sets (Cover & Thomas, 2006). In information theory, the $\varepsilon$-typical set $A_\varepsilon^{(n)}$ for a distribution $p$ over sequences $x^n$ of length $n$ (whether i.i.d. or a stationary Markov chain), with entropy rate $H$, is defined as

$$A_\varepsilon^{(n)} = \left\{ x^n : H - \varepsilon \leq \frac{-\log p(x^n)}{n} \leq H + \varepsilon \right\}. \quad (3)$$

The quantity $\frac{-\log p(x^n)}{n}$ is the negative log-likelihood (NLL)

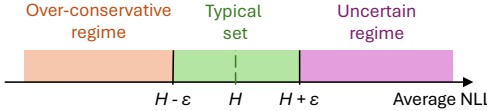

*Figure 4.* Different regimes related to the typical set.

of the sequence $x^n$ normalized by the sequence length $n$, referred to as the average NLL. Intuitively, typical sets capture the bulk of probability mass in a distribution, and a sequence is "typical" if its NLL is near the true entropy rate. Checking whether the average NLL lies within an $\varepsilon$-band around $H$ amounts to verifying whether $x^n$ belongs to the typical set, where a smaller NLL means a more likely sequence, and vice versa.

When we train probe-LLM on data drawn from the Markov chain and then evaluate the sequences generated by the trained model under the *same* Markov chain, we make the following interpretation:

- If probe-LLM generates a sequence whose average NLL (under the Markov chain) is below the typical set's lower bound, then the sequence is too likely. This suggests that the LLM may be overly conservative, producing highly predictable or repetitive sequences.
- If probe-LLM generates a sequence whose average NLL is above the typical set's upper bound, the sequence is too unlikely. This includes scenarios where the LLM is generating patterns that deviate significantly from its training distribution.

Hence, typical sets provide a principled way to detect whether sequences from a trained model are "too predictable" or "too imaginative" relative to the true data distribution. Extending beyond the standard typical set concept, we can define the two regimes outside the typical set with LLM-specific meanings, namely, "over-conservative" and "uncertain," as shown in Figure 4. This definition of regimes can help us further understand and analyze LLM behaviors using this theoretical tool.

**Applying Data Probes to LLM Training.** To put data probes into practice, we ran a preliminary experiment with the target entropy rate $H = 1$ bit/token and Markov chain state space size (equivalent to the token vocabulary size)

*Table 2.* Exemplar generated output (first 128 tokens) of GPT-2 small model (see Appendix E for the full non-truncated result). The table reports probe-side quantitative diagnostics and qualitative real-side alignment.

| Method | GPT-2 trained on data probes (**probe-LLM**)
Prompt: 1 | Pre-trained GPT-2 on real data (**text-LLM**)
Prompt: Tell me about machine learning. |
|---|---|---|
| Greedy | 1, 5, 127, 117, 99, 61, 5, 127, 117, 99, 61, 5,
127, 117, 99, 61, 5, 127, 117, 99, 61, 5,
...
*(Average NLL:* 0.694, *in over-conservative regime)* | Machine learning is a new field of research that has been around for a while.
It's a new field of research that has been around for a while. ...
*(Over-conservative, repetitive text without much information)* |
| Sampling, $T = 1$ | 1, 5, 127, 117, 99, 88, 41, 56, 35, 29, 109, 65, ...

*(Average NLL:* 0.866, *in typical set)* | Machine learning is one of the most popular applications ... |
| Sampling, $T = 1.3$ | 1, 5, 78, 90, 35, 29, 109, 79, 16, 11, 51, 107, ...

*(Average NLL:* 0.979, *in typical set)* | I've been trying for the past 20 years to develop ways ... |
| Sampling, $T = 1.5$ | 1, 5, 78, 90, 35, 29, 7, 7, 80, 35, 29, 28, ...

*(Average NLL:* 1.406, *in uncertain regime)* | If I get an education from a company on your work project ...

*(Uncertain, not closely related to the prompted topic of machine learning)* |

**Claim Card for Table 2.**
**Claim:** Increasing temperature increases average NLL and moves probe outputs from over-conservative toward uncertain regimes under the known Markov law.
**Intervention:** Temperature contrasts $T \in \{0, 1.0, 1.3, 1.5\}$.
**Probe-side diagnostics:** Average NLL and typical-set regime labels.
**Real-side connection:** Qualitative alignment of degeneration and diversity behavior in text outputs.
**Pre-declared failure condition:** Predicted probe-side directional regime movement is not observed.
**Transfer status for this table:** Preliminary controlled validation step with qualitative alignment, not a formal transfer verdict.

$M = 128$. Using the sequences generated by the Markov chain, we train our probe-LLM. Because the data is synthetic, it can be scaled without concern for scarcity, and a test set can easily be generated from the same Markov chain. We use a sequence length of $n = 128$ for both the training and test data. After training, we generate new sequences from the LLM using either greedy decoding, which always picks the most probable next token, or sampling with a specified *temperature* $T$. Higher $T$ encourages diversity, and vice versa, while greedy decoding gives the lowest diversity.

**Analyzing Practical Observations through the Lens of Typical Sets.** Table 2 shows some examples of generated data from both probe-LLM (after training) and text-LLM. We observe that *both models show qualitatively similar decoding behaviors*. When choosing $\varepsilon = 0.2$ in the typical set definition, the main observations are as follows: For this experiment, the testable claim is that increasing temperature should increase probe-side average NLL and move probe outputs from over-conservative regimes toward uncertain regimes under the known Markov distribution, with corresponding qualitative shifts in real-text outputs. The pre-declared failure condition is that this probe-side directional pattern is not observed. For real-text outputs in Table 2, we report qualitative alignment only. This is a preliminary controlled validation step and does not by itself establish a formal transfer verdict.

1. Greedy sampling generates repetitive outputs from both models, where for probe-LLM the generated sequence is in the over-conservative regime outside the typical set.
2. Sampling with temperatures $T = 1$ and $T = 1.3$ from probe-LLM gives sequences within the typical set, as seen by their average NLL values. With the same sampling, text-LLM generates meaningful outputs relevant to the input prompt as well.
3. Sampling with temperature $T = 1.5$, both models gen-

erate outputs that are mostly irrelevant to the inputs, as indicated by the text output and average NLL value from text-LLM and probe-LLM, respectively.

These findings verify our earlier discussion on different ranges of average NLL values and their relation to the typical set and its adjacent regimes.

We further examine the average NLL distribution of sequences generated by probe-LLM, with each prompt being one of the Markov chain states following the stationary distribution $\pi$, as shown in Figure 5. We have some interesting observations as follows:

1. Compared to the average NLL distribution of sequences sampled from the ground-truth Markov chain (blue curve in Figure 5), both greedy decoding and sampling with $T = 1$ tend to generate sequences with higher likelihood values. This is an interesting finding because the loss function used in training has a mathematically equivalent form as sampling with $T = 1$. However, when using the trained model for generating long sequences (127 tokens from one input token in our experiment), the distribution of the generated sequence departs from that of the original Markov chain. *This aligns with the practical experience that LLMs tend to generate less useful content when the amount of generated content is large compared to the input prompt.*
2. When sampling with $T = 1.25$, the model tends to generate sequences with slightly higher likelihood compared to the ground-truth Markov chain, but there are also some instances where the model generates very unlikely sequences (i.e., high NLL). *This matches the experience that LLMs tend to generate more similar content than humans while hallucinating at times* (Ye et al., 2024).

In all the above results, *the computation of NLL has been only possible because we know the ground-truth distribution* that is specified by the Markov chain. It is particularly

*Table 3.* Comparison of representative prior studies against data-probe criteria. $C_1$: known process fully defined and samplable. $C_2$: controllable and interpretable intervention knobs. $C_3$: diagnostic metrics computable. $C_4$: pre-declared falsification/transfer criteria.

| Theme & representative existing research | Data-probe criterion satisfaction | | | | Probe-based study to close gaps |
|---|---|---|---|---|---|
| | $C_1$ | $C_2$ | $C_3$ | $C_4$ | |
| Data diversity, sufficiency & complexity (e.g., Makkuva et al. (2025); Rajaraman et al. (2024)) | Yes | Partial | Yes | No | Declare intervention knobs and contrast grids ($C_2$). Add pre-registered failure rules and transfer-status labels per contrast ($C_4$). |
| Overfitting, regularization & data curation (e.g., Wettig et al. (2024); Penedo et al. (2024)) | No | Partial | Yes | No | Introduce known-process probe generators with sampling laws ($C_1$). Specify intervention knobs and ranges ($C_2$). Add pre-declared falsification and IV/EV verdict reporting ($C_4$). |
| Adaptation, transfer & in-context learning (e.g., Von Oswald et al. (2023); Edelman et al. (2024)) | Partial | Yes | Yes | No | Map shifts to source-process assumptions whenever possible ($C_1$). Add pre-registered external-validity failure criteria and transfer labels ($C_4$). |
| Robustness and adversarial testing (e.g., Sainz et al. (2023); Shu & Yu (2024)) | No | Partial | Yes | No | Define reduced known-process stress generators ($C_1$). Declare perturbation knobs and strengths ($C_2$). Attach falsification thresholds and IV/EV verdicts ($C_4$). |
| Information-theoretic understanding of LLMs (e.g., Zekri et al. (2024)) | Yes | Partial | Yes | No | Standardize intervention knobs for information-theoretic contrasts ($C_2$). Add pre-declared transfer rejection rules and status labels ($C_4$). |
| Mechanistic interpretability and inner-model analysis (e.g., Singh et al. (2024); Räuker et al. (2023)) | No | Partial | Partial | No | Use known structured process families for targeted mechanisms ($C_1$). Declare intervention knobs over data factors ($C_2$). Define computable diagnostics tied to those factors ($C_3$). Add falsification predicates and transfer criteria ($C_4$). |

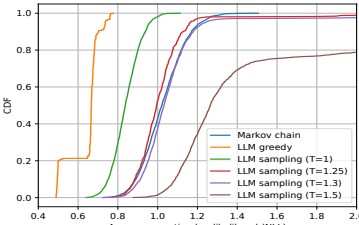

*Figure 5.* Cumulative density function (CDF) of average NLL of generated sequences.

interesting that from the average NLL results and their connection to the typical set concept, we are able to observe important LLM behaviors seen in practice from the simple GPT-2 model trained using data probes. This illustrates the potential of data probes for achieving an in-depth understanding and analysis of LLMs, thus **we advocate for a systematic development of data probes and their uses.**

## 5. Potential Problems That Can Be Studied Using Data Probes

We now discuss some specific sets of problems that can potentially be tackled using data probes, which showcase the flexibility of data probes as tools for both theoretical insights and practical improvements in LLM development. Table 3 provides a detailed comparison between representative prior studies and the four formal criteria $C_1$–$C_4$, and the open problems for a probe-based study.

**Data Diversity, Sufficiency, and Complexity.** A fundamental question in LLM research is: *How much data is truly needed for effective training or fine-tuning?* By systematically varying the properties of data probes, one can precisely explore the following metrics.

*Entropy.* Adjusting entropy parameters allows researchers to investigate when a model begins to underfit or overfit. Experiments can determine the minimum dataset size (i.e., number of sequences or tokens) required for a given level of performance, thereby revealing data-sufficiency thresholds.

*Vocabulary Size.* Generating data probes with different numbers of possible tokens can reveal how vocabulary size affects the learning, generalization, and overfitting.

*Dependency Structure.* Designing data probes with multi-order dependencies helps analyze the model's capacity to capture complex long-range correlations. By comparing the performance of different architectural choices or hyperparameters, one can isolate which aspects of the model contribute most to handling higher-order dependencies.

*Probabilistic Context-Free Grammar (PCFG) Probes.* A concrete extension is to use PCFG-style probes with knobs for maximum tree depth, branching-factor distribution, and rule entropy/imbalance (Manning & Schutze, 1999). Under fixed intervention contrasts, diagnostics can include depth-conditioned failure rates, shifts in likelihood-based regime occupancy, and robustness of behavior under controlled grammar perturbations. This creates a tractable complexity ladder from Markov-style probes to richer hierarchical probes while preserving interpretable control variables.

*Connection to Existing Studies.* Controlled Markov or structured synthetic settings isolate important parts of this question (Makkuva et al., 2025; Rajaraman et al., 2024; Zekri et al., 2024), but they are typically not framed with pre-declared falsification and transfer-status reporting.

**Overfitting, Regularization, and Data Curation.** With a fully known data probe distribution, researchers can exactly measure the gap between the model's learned distribution and the true distribution and investigate into the following.

*Comparing Regularizers.* Investigations of weight decay, dropout, layer norm, or more sophisticated techniques can be done under identical data conditions, allowing precise quantification of each method's impact on overfitting.

*Data Filtering Strategies.* One can introduce "low-quality" tokens or sequences into the data probes, then apply different filters to remove these artifacts before training. This helps identify which filtering strategies effectively balance

data diversity with cleanliness and how they impact LLM performance on typical and out-of-distribution sequences.

*Imbalanced Data Probes.* Generating controlled imbalances in token frequencies or sequence types allows the investigation of how reweighting, oversampling, or undersampling may mitigate bias or improve generalization.

*Connection to Existing Studies.* Current data filtering and curation studies provide strong empirical guidance (Wettig et al., 2024; Penedo et al., 2024; Su et al., 2025; Gohari et al., 2026), and the probe methodology reframes them with intervention knobs, pre-registered failure conditions, and matched transfer diagnostics.

**Adaptation, Transfer, and In-Context Learning.** Real-world deployment often requires adapting an LLM to new domains or shifting data distributions. Data probes can replicate such scenarios by 1) generating an initial "base" distribution $\mathcal{D}_{\text{base}}$ for model training, and 2) constructing a related "adaptation" distribution $\mathcal{D}_{\text{adapt}}$ that differs in entropy, vocabulary composition, or correlation patterns. Researchers can then examine important stages in an LLM workflow.

*Fine-Tuning Behavior.* How quickly and effectively does the LLM adapt from $\mathcal{D}_{\text{base}}$ to $\mathcal{D}_{\text{adapt}}$? Do certain model architectures or algorithms enable a smoother transition?

*In-Context Learning.* Instead of fine-tuning on $\mathcal{D}_{\text{adapt}}$, one may supply a small sample of the new distribution via input prompts, without updating model parameters. This setup reveals how the gap between $\mathcal{D}_{\text{base}}$ and $\mathcal{D}_{\text{adapt}}$ influences the model's in-context learning capability.

*Connection to Existing Studies.* In-context-learning theory and synthetic analyses show how controlled distributions can reveal adaptation behavior (Von Oswald et al., 2023; Edelman et al., 2024). Probe redesign adds claim cards and pre-declared transfer verdicts for each shift scenario.

**Robustness and Adversarial Testing.** Data probes can be designed to stress-test an LLM's resilience.

*Noisy or High-Entropy Perturbations.* By injecting random tokens or increasing the likelihood of rare transitions, one can measure if the LLM under- or overestimates rare events and how it handles sudden distributional changes.

*Adversarial Sequences.* Data probes can be engineered to systematically push the model to produce incorrect or inconsistent token predictions, thus exposing potential weaknesses of the model.

*Connection to Existing Studies.* Robustness and contamination work identifies major empirical failure modes (Sainz et al., 2023; Shu & Yu, 2024). Probe-based studies would make perturbation strength, preserved statistics, and falsification thresholds clear.

**Towards an Information-Theoretic Understanding of LLMs.** One of the most promising aspects of data probes is their alignment with formal analytical methods. Since the distribution is explicitly specified, classical information-theoretic tools become directly applicable.

*Typical Sets and Entropy Bounds.* Tracking how many sequences generated by the LLM fall within certain NLL bounds provides a way to assess whether the model is over-conservative or unlikely.

*Scaling Laws.* As entropy or vocabulary size grows, one can characterize how model capacity needs to scale to maintain a given level of perplexity or generalization.

*Transformer Analysis with Controlled Input.* Simplified or theoretical variants of transformer architectures can be analyzed when driven by data probes. This offers a path toward an "information theory for LLMs," linking capacity, data complexity, and phenomena observed in large models.

*Connection to Existing Studies.* This builds on the current "transformers on controlled distributions" line (Makkuva et al., 2025; Rajaraman et al., 2024; Zekri et al., 2024) by adding uniform validity predicates and pre-declared transfer decision rules across experiments.

**Mechanistic Interpretability and Controlled-Data Analysis.** Data probes can catalyze interpretability research, since the ground-truth data generation process is explicitly known.

*Disentangling Neural Circuits.* One can identify which internal components, e.g., attention heads, neurons, or entire submodules, respond to specific statistical patterns in the data probe without influence from hidden real-world biases.

*Identifying Emergent Behaviors.* If higher-order dependencies are introduced intentionally, the discovery of "emergent circuits" for handling those dependencies can be intuitively linked to the data probe's known structure.

*Connection to Existing Studies.* Mechanistic interpretability provides taxonomies and candidate internal explanations (Singh et al., 2024; Räuker et al., 2023), while probe methodology adds controlled causal attribution from data factors to those internal mechanisms.

## 6. Alternative Views

Despite the potential advantages of data probes, some may argue that synthetic data is too far from real-world language to offer meaningful insights. The carefully controlled distributions in data probes often omit cultural, contextual, and factual dimensions that LLMs need to handle in practice. One may question whether results derived from clean artificial distributions will transfer to domains where text is messy or influenced by subtle socio-linguistic factors.

A related concern is that, in focusing on theoretical properties such as entropy or typical sets, data probes may prioritize purely statistical characteristics over more complex linguistic phenomena like compositionality, pragmatics, or style. Exploring these richer aspects of language may demand large-scale real-world corpora, which are far beyond

any synthetic distribution one could design.

However, even if data probes are admittedly simplified, there is value in seeking a common exploratory tool that bridges theoretical investigation and empirical testing. Real-world language is inherently complicated, and isolating particular factors that affect model performance can be challenging when multiple confounders coexist. Data probes, in contrast, allow for the controlled manipulation of key distributional features, enabling researchers to attribute changes in LLM behavior to specific data properties. Results from such controlled experiments can still guide subsequent analyses on authentic datasets. Indeed, insights gleaned from data probes, such as detecting over-conservatism or uncertainty based on typical-set bounds, can inform how we design data filters, evaluate model outputs, and interpret perplexity metrics in real-world scenarios.

## 7. Further Discussions

**From Empirical to Foundational Approaches.** Many scientific fields have evolved from primarily empirical practices, where progress hinges on large-scale observations or experiments, to more foundational theory-driven frameworks that formalize and explain the phenomena being studied. In the case of LLMs, advances have so far been relying largely on massive datasets and benchmark performance. Although this empirical strategy has yielded remarkable capabilities, a deeper theoretical understanding on how and why certain methods work remains open. Therefore, the concept of data probes is an essential but also natural step when seen from the perspective of science evolution throughout the history. It is also worth mentioning that, even in as early as 1948 (Shannon, 1948), Shannon mentioned that "a sufficiently complex stochastic process will give a satisfactory representation of a discrete source," which to some degree foresaw the importance and rise of data probes.

**Perspectives on Knowledge and Creativity.** An open question is whether factual knowledge and genuine creativity can both be simulated by data probes generated from stochastic processes. Facts are often viewed as information that can be encoded and reproduced by a sufficiently capable system, while creativity implies novel associations or the introduction of entirely new ideas that may not be directly derivable from any fixed distribution. A possibility of simulating creativity with data probes is to have specifically tailored "creative" sequences which are derived from a separate stochastic function that takes the "non-creative" data probes or their distribution as input. As we push the boundaries of AI, experiments with such "creative" data probes may be key to understanding where algorithmic ingenuity ends and authentic insight begins.

**Future Research.** While our example in Section 4 uses relatively basic generative mechanisms and experiments, future research should aim at capturing richer aspects of language

and pushing the data-probe approach towards practical adoption. Possible research avenues include the following.

- Develop a family of *powerful data probes* for studying various classes of important problems.
- Incorporate *hierarchical structures* into data probes with nested or layered rules, such as those found in syntax or multi-level dependencies, to test an LLM's capacity for compositional reasoning.
- Develop data probes with *contextual variants*, including context-specific constraints or dynamic scenarios in which the valid transitions depend on earlier tokens in more intricate ways.
- Simulate the effect parallel data in *multiple languages or other data modalities* using data probes, enabling controlled studies of LLMs in more complex domains.
- *Integrate* data-probe generation mechanisms and experimentation *into practical LLM workflows*.

**Call to Action.** We call for a closer collaboration between theoretical and empirical research communities through the shared development and use of data probes. Theorists can design data probes with explicit statistical and information-theoretic structure that enable formal analysis, while empirical researchers can deploy these probes at scale to test whether theoretical predictions manifest in practical models and workflows. Industry practitioners can further strengthen this connection by integrating data probes into training, fine-tuning, and evaluation pipelines as controlled diagnostics that complement real-world datasets. To support this convergence, funding agencies and institutions should prioritize shared benchmarks, open libraries, and collaborative venues centered on data-probe methodologies. By establishing data probes as a common experimental interface, the community can bridge long-standing divides between theory and practice and move toward a more principled understanding of how data influences LLM behavior.

## 8. Conclusion

Data probes, as introduced in this paper, offer a resource-efficient and theory-friendly approach for studying how LLMs behave under well-controlled conditions. Although they do not attempt to replicate the full complexity of real-world data, data probes enable an important exploratory step, complementing large-scale empirical data and allowing for both targeted experimentation and deeper theoretical insights. By systematically varying factors in the data-probe generation process, researchers can gain a clearer picture of model capabilities and limitations before scaling up to massive corpora. We therefore advocate for developing mechanisms for generating powerful data probes that have high theoretical and practical values simultaneously and integrate such mechanisms into mainstream LLM workflows for practical adoption.

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

# Appendix

## A. Impact Statement

The use of data probes, as presented in this paper, can substantially lower the entry barrier for studying LLMs by reducing the need for massive datasets and extensive computational resources. This democratization of LLM research has the potential to broaden participation in AI development, enabling a wider range of academic institutions, smaller companies, and independent researchers to investigate critical issues such as model robustness, fairness, and interpretability. From an ethical standpoint, data probes provide a controllable test bed for examining undesirable behaviors, such as hallucinations or overfitting to sensitive attributes, without requiring access to potentially harmful or privacy-sensitive real-world text corpora. By systematically isolating the factors that lead to biased or misleading outputs, researchers and practitioners can preemptively identify failure modes before deploying models in real-world applications. This helps mitigate risks associated with biased or misleading AI-generated content, ultimately contributing to more responsible AI development.

## B. Formal Definition with Extended Notation

This appendix provides the complete formal definition.

**A.1 Formal Object.** Let $\mathcal{X}$ be the output space and let $\mathcal{E}$ be the set of valid configurations of a known generative process. For each $\eta \in \mathcal{E}$, let $p_\eta$ denote the distribution on $\mathcal{X}$ defined by configuration $\eta$. We define
$$\Pi = (\mathcal{P}, \mathcal{M}, \mathcal{H}, \mathcal{F}), \tag{4}$$
where $\mathcal{P} = (\{p_\eta\}_{\eta \in \mathcal{E}}, \mathcal{U}, \mathcal{V}, \mathcal{I})$, with $\mathcal{U}$ a finite set of interpretable control knobs, $\mathcal{V} = \{\mathcal{V}_u\}_{u \in \mathcal{U}}$ the admissible values, and $\mathcal{I} = \{I_{u,v} : \mathcal{E} \to \mathcal{E}\}$ the intervention family.

Let $\mathcal{M} = \{m_j\}_{j=1}^J$ be diagnostics with each $m_j : \mathcal{Y} \to \mathbb{R}$ for diagnostic-object space $\mathcal{Y}$. Let $\mathcal{H}$ be claims of the form $h = (u, v_a, v_b, m_j)$ with $u \in \mathcal{U}$, $v_a, v_b \in \mathcal{V}_u$, and $m_j \in \mathcal{M}$. Let $\mathcal{F} = (B, \mathcal{R}, \alpha)$ where $B(h) = (s_h, \delta_h, \mathcal{R}_h)$ binds expected direction $s_h \in \{-1, +1\}$, probe-side margin $\delta_h \geq 0$, and assigned falsification predicates $\mathcal{R}_h \subseteq \mathcal{R}$, and where $\alpha \in (0, 1)$ is a pre-registered significance level.

**A.2 Validity Predicates.** The four methodological criteria are made operational with:

- $C_1(\Pi)$: each $p_\eta$ is fully specified and samplable for every $\eta \in \mathcal{E}$,
- $C_2(\Pi)$: each knob $u \in \mathcal{U}$ has admissible set $\mathcal{V}_u$ and each intervention map $I_{u,v}$ is defined,
- $C_3(\Pi)$: each diagnostic $m_j \in \mathcal{M}$ is computable on $\mathcal{Y}$,
- $C_4(\Pi)$: each claim $h \in \mathcal{H}$ has nonempty assigned falsification set $\mathcal{R}_h$ through $B(h)$.

The definition is valid iff $C_1(\Pi) \wedge C_2(\Pi) \wedge C_3(\Pi) \wedge C_4(\Pi)$.

**A.3 Two-Layer Evaluation.** Fix baseline configuration $\eta^0 \in \mathcal{E}$ and claim $h = (u, v_a, v_b, m_j)$ with $B(h) =$

$(s_h, \delta_h, \mathcal{R}_h)$. Define intervened configurations $\eta^a = I_{u,v_a}(\eta^0)$ and $\eta^b = I_{u,v_b}(\eta^0)$. Define probe contrast
$$\Delta_h^{\text{probe}} = \mu_b^{\text{probe}} - \mu_a^{\text{probe}}, \tag{5}$$
$$\mu_a^{\text{probe}} = \mathbb{E}_{Y \sim p_{\eta^a}}[m_j(Y)], \tag{6}$$
$$\mu_b^{\text{probe}} = \mathbb{E}_{Y \sim p_{\eta^b}}[m_j(Y)]. \tag{7}$$

For matched practical evaluation, define real-side contrast $\Delta_h^{\text{real}} = \mu_b^{\text{real}} - \mu_a^{\text{real}}$ using matched intervention settings and matched diagnostic. Let $z_h^{\text{probe}}$ and $z_h^{\text{real}}$ denote statistical summaries used by falsification predicates (for example one-sided p-values, confidence-interval bounds, and seed-stability indicators).

We use hypothesis testing and stability checks via predicates in $\mathcal{R}_h$:
$\text{IV}(h) = 1$
$$\iff s_h \Delta_h^{\text{probe}} \geq \delta_h \wedge \forall r \in \mathcal{R}_h, \, r(\text{probe}, z_h^{\text{probe}}, \alpha) = 1, \tag{8}$$

$\text{EV}(h) = 1$
$$\iff s_h \Delta_h^{\text{real}} \geq 0 \wedge \forall r \in \mathcal{R}_h, \, r(\text{real}, z_h^{\text{real}}, \alpha) = 1. \tag{9}$$

Transfer is accepted iff $\text{IV}(h) = 1 \wedge \text{EV}(h) = 1$. If $\text{IV}(h) = 1$ but $\text{EV}(h) = 0$, the claim is probe-local by definition.

**A.4 Bottom-Up and Top-Down Usage.** Bottom-up starts from a theoretical mechanism and defines claims under this same protocol. Top-down starts from an observed practical failure and constructs a reduced process specification with documented removed factors and preserved properties. Both directions use the same object $\Pi$, the same validity predicates, and the same transfer decision rule.

**A.5 Mapping to the Current Example.** In the current basic example (Section 4.3), the process family is Markov and the control knob is temperature with values $\{0, 1.0, 1.3, 1.5\}$ where $0$ corresponds to greedy decoding. Claims are temperature-pair contrasts with fixed process configuration and fixed probe checkpoint. Diagnostics are average NLL and typical-set regime labels. Under this protocol, the table in the main text is a preliminary controlled validation step with qualitative real-side alignment, while formal transfer requires explicit external-validity pass criteria.

## C. More Details on Experiments

For the probe-LLM, we used a Markov chain with 128 states, selected to have an entropy rate near 1 bit/token. We trained a GPT-2–like architecture from scratch, with a reduced vocabulary of 128 tokens. We used AdamW with a learning rate of $10^{-5}$, weight decay of 0.01, batch size of 4, and train for 10,000 steps on one NVIDIA RTX 3090 GPU. This number of training steps is sufficient for convergence, as we observed in our experiments. The process of generating

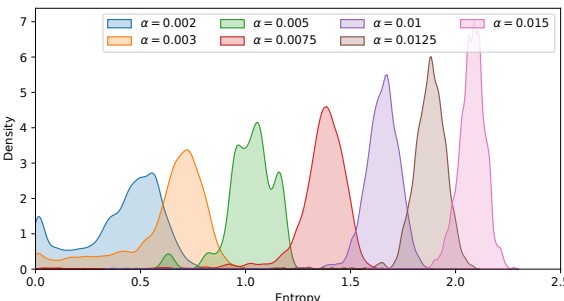

*Figure 6.* Entropy value distribution of randomly generated Markov Chains (128 states) with Dirichlet parameter $\alpha$.

synthetic data required negligible storage or curation. This setup's simplicity highlights how one can conduct controlled LLM experiments without the large overhead of real-text pipelines.

## D. Generating a Markov Chain with Target Entropy Rate

Let us define $M$ as the number of states in the Markov chain, which is equivalent to the vocabulary size of the tokens used with probe-LLM. To create our Markov chain, we sample each row of a transition matrix from a Dirichlet distribution Dirichlet$(\alpha, \alpha, \ldots, \alpha)$, where the parameter $\alpha > 0$ is the same across all the $M$ dimensions and controls the variability of the transition probabilities. We generate multiple candidate transition matrices in this manner.

After sampling candidate transition matrices, we compute the entropy rate of each Markov chain. For a Markov chain with transition matrix $P$ and stationary distribution $\pi$, the entropy rate $H(P)$ is given by

$$H(P) = -\sum_{i=1}^{M} \pi_i \sum_{j=1}^{M} P_{ij} \log P_{ij}. \quad (10)$$

We select the transition matrix $P^*$ whose entropy rate $H(P^*)$ is closest to a specified target $H$. Figure 6 shows that the randomly generated transition matrices give broad ranges of entropy values, especially when using different values of $\alpha$, based on which a transition matrix with an entropy close to the desired value $H$ can be easily selected.

Once a transition matrix $P^*$ is selected, we generate token sequences by sampling from the Markov chain with transition matrix $P^*$. These sequences form our data probes for training probe-LLM. Because the Markov chain is fully specified, we can generate an effectively unlimited amount of data aligned with the known distribution governed by $P^*$.

## E. Exemplar Generated Output

The full results of Table 2 is shown in Table 4.

*Table 4.* Exemplar generated output (first 128 tokens) of GPT-2 small model.

| Method | GPT-2 trained on probe data (**probe-LLM**)
Prompt: 1 | Pre-trained GPT-2 on real data (**text-LLM**)
Prompt: `Tell me about machine learning.` |
|---|---|---|
| Greedy | `1, 5, 127, 117, 99, 61, 5, 127, 117, 99,`
`61, 5,`
`127, 117, 99, 61, 5, 127, 117, 99, 61, 5,`
`127, 117, 99, 61, 5, 127, 117, 99, 61, 5,`
`127, 117, 99, 61, 5, 127, 117, 99, 61, 5,`
`127, 117, 99, 61, 5, 127, 117, 99, 61, 5,`
`127, 117, 99, 61, 5, 127, 117, 99, 61, 5,`
`127, 117, 99, 61, 5, 127, 117, 99, 61, 5,`
`127, 117, 99, 61, 5, 127, 117, 99, 61, 5,`
`127, 117, 99, 61, 5, 127, 117, 99, 61, 5,`
`127, 117, 99, 61, 5, 127, 117, 99, 61, 5,`
`127, 117, 99, 61, 5, 127, 117, 99, 61, 5,`
`127, 117, 99, 61, 5, 127`

*(Average NLL:* 0.694, *in over-conservative regime)* | `Machine learning is a new field of research that has`
`been around for a while.`
`It's a new field of research that has been around for a`
`while.`
`It's a new field of research that has been around for a`
`while.`
`It's a new field of research that has been around for a`
`while.`
`It's a new field of research that has been around for a`
`while.`
`It's a new field of research that has been around for a`
`while.`
`It's a new field of research that has been around for a`
`while.`
`It's a new field of research that has been around for a`
`while.`
`It's a new field of research that has been around for a`
`while`

*(Over-conservative, repetitive text without much information)* |
| Sampling,
$T = 1$ | `1, 5, 127, 117, 99, 88, 41, 56, 35, 29,`
`109, 65, 68, 14, 6, 8, 92, 31, 22, 122, 70,`
`25, 38, 47, 42, 49, 20, 29, 109, 79, 16,`
`55, 37, 42, 49, 20, 29, 109, 65, 127, 117,`
`99, 61, 5, 127, 117, 99, 61, 69, 64, 79,`
`16, 11, 51, 123, 51, 123, 41, 56, 19, 35,`
`29, 109, 65, 127, 117, 99, 61, 5, 127, 14,`
`6, 8, 92, 31, 22, 122, 70, 37, 42, 49, 110,`
`63, 32, 122, 70, 25, 38, 47, 42, 49, 110,`
`63, 35, 29, 109, 65, 68, 14, 6, 58, 93, 99,`
`61, 68, 53, 30, 93, 99, 61, 68, 14, 6, 58,`
`93, 99, 61, 5, 127, 117, 99, 88, 41, 56,`
`35, 29, 109, 79`

*(Average NLL:* 0.866, *in typical set)* | `Machine learning is one of the most popular`
`applications for solving problems. It has been used`
`successfully in many industries such as health care`
`and financial services, education and government.`
`There is currently a great deal of excitement about`
`its usefulness. The technology can be used to predict`
`the future for real time.`

`So what is the value of machine learning?`

`First, as an exercise, let's imagine a case where we`
`are looking at an average daily life of a person. We`
`are looking at our average life span in months. Our`
`average age is around 30 and the average lifetime`
`length of time is` |
| Sampling,
$T = 1.3$ | `1, 5, 78, 90, 35, 29, 109, 79, 16, 11, 51,`
`107, 0, 78, 90, 35, 29, 109, 65, 127, 117,`
`99, 88, 41, 56, 91, 70, 88, 41, 56, 35, 29,`
`109, 79, 81, 89, 3, 103, 13, 23, 85, 121,`
`62, 96, 75, 82, 111, 15, 33, 7, 11, 51,`
`123, 90, 35, 29, 109, 65, 127, 117, 99, 61,`
`5, 127, 117, 58, 83, 62, 1, 5, 127, 117,`
`99, 88, 41, 56, 91, 70, 25, 38, 47, 42, 49,`
`66, 12, 125, 86, 14, 6, 58, 83, 62, 116,`
`95, 124, 43, 8, 13, 23, 85, 121, 63, 32,`
`122, 70, 37, 42, 49, 110, 85, 121, 43, 8,`
`92, 116, 13, 0, 78, 90, 35, 29, 109, 79,`
`16, 11, 51, 107, 24`

*(Average NLL:* 0.979, *in typical set)* | `I've been trying for the past 20 years to develop ways`
`to implement machine learning models like Deep Neural`
`Networks, Convolutional Neural Networks, Autoencoders,`
`and other general-purpose networks like Deep Learning`
`Neural Networks. My initial goals are to do what I'd`
`call a "learning machine", which I call the general`
`purpose neural network. I was starting off with a`
`single neuron (the top 10% of a population), and`
`eventually decided the data needed were only a small`
`portion of human-related data: about 5% to about 5% of`
`that population! So it's basically a two-process` |
| Sampling,
$T = 1.5$ | `1, 5, 78, 90, 35, 29, 7, 7, 80, 35, 29, 28,`
`90, 35, 29, 109, 65, 115, 56, 84, 12, 84,`
`12, 84, 12, 125, 124, 43, 8, 92, 116, 95,`
`28, 126, 9, 14, 6, 8, 92, 106, 21, 25, 38,`
`47, 42, 70, 37, 42, 22, 122, 70, 37, 42,`
`49, 110, 99, 61, 68, 62, 116, 95, 124, 97,`
`21, 25, 38, 47, 42, 49, 110, 63, 32, 122,`
`74, 95, 124, 120, 29, 28, 90, 116, 95, 13,`
`23, 85, 121, 62, 1, 5, 127, 14, 6, 42, 49,`
`62, 116, 13, 94, 127, 117, 58, 83, 62, 96,`
`75, 82, 111, 15, 2, 21, 60, 108, 21, 25,`
`38, 47, 42, 5, 127, 117, 58, 93, 99, 88,`
`41, 56, 91, 70`

*(Average NLL:* 1.406, *in uncertain regime)* | `If I get an education from a company on your work`
`project that's interesting, is interesting because this`
`guy will find all this fascinating stuff so they will`
`create other interesting stuff and I think so on. This`
`will be pretty big and I feel like I'm probably going`
`to go back to a simpler version.`

`QQ What makes machines better than real people? If an`
`expert gets a grant that has some interesting results,`
`if you get to actually have it analyzed, there's not`
`much difference at all in the way human minds think`
`in situations where people make assumptions on whether`
`things aren't interesting, how are`

*(Uncertain, not closely related to the prompted topic of machine learning)* |

