# OpenReview forum: "Position: Let's Develop Data Probes to Fundamentally Understand How Data Affects LLM Performance"
_ICML.cc/2026/Position_Paper_Track — ICML 2026 Position Paper Track regular_

### Official Review · Reviewer_rLEz · 2026-02-28

**Significance:** 2
**Argument Clarity:** 3
**Rating:** 4
**Confidence:** 4

**Questions:**

Q1. How to define and evaluate such probe? i.e., Is there any formal definition of data probe, and how can  we evaluate the  effectiveness of this probe?

Q2. What is the precise relationship and differentiation between data probes and prior work on synthetic/toy tasks and formal language benchmarks?

**Alternative Views Section:**

Yes

**Compliance With Llm Reviewing Policy A Conservative:**

Affirmed.

**Discussion Potential:**

3

**Ethical Review Concerns:**

Not applicable.

**Final Justification:**

The authors have provided additional clarifications to address the concerns during reviewing period, thus improving the  score accordingly.

**Paper Summary:**

This position paper proposes data probes, that synthetic sequences sampled from fully known stochastic processes to study how data properties influence LLM training and decoding. Because the generating distribution is known, probes allow exact computation of likelihood and unlimited data generation. This paper also argues probes can bridge simplified theory and costly empirical work.

**Position:**

Yes

**Position In Title:**

Yes

**Related Work:**

2

**Strengths And Weaknesses:**

**Strengths**

1. This paper’s use of controllable synthetic data from known distributions to isolate variables and study causal links from data properties to model behaviors is a reasonable and well-motivated direction.

2. The known generating distribution allows the likelihood and other statistical indicators to be computed precisely, which is an important advantage over natural language corpora because it enables clearer and more diagnostic analyses.

3. This proposal, even if not yet fully mature, is likely to spark constructive discussion about what constitutes a good probe and which findings can transfer to natural-language settings.

**Weaknesses**

1. This paper claims that probes can bridge theory and practice, but the current version lacks an operational research agenda, e.g., principled probe design guidelines.

2. The example mainly reproduces commonplace decoding phenomena e.g., greedy repetition and higher temperature leading to more off-topic generations. But it does not clearly articulate what new diagnostic signal data probes provide compared to more standard analyses or benchmarks.

**Support:**

2

---

> ### Author Rebuttal · Authors · 2026-03-30
>
> We thank the reviewer for the thorough feedback and for acknowledging that our paper can spark constructive discussions.
>
> **W1 (Operational research agenda).**
> The data-probe methodology provides an operational research agenda that links theory-facing analysis and practice-facing diagnosis through one shared protocol.  In the bottom-up direction, controlled probes produce handles that can be checked in practical LLM settings. In the top-down direction, observed practical failures are reduced into tractable probe families with documented simplifications. Both directions are supported by the data-probe methodology.
> This methodology also requires four criteria that every probe must satisfy (and where we advocate for focusing research efforts):
> - a known generative process,
> - controllable and interpretable knobs,
> - a measurable main metric, and
> - a testable and falsifiable claim with a pre-declared failure condition.
>
> These criteria support hypothesis-driven research with rigorous validation and keep theory and practice aligned through checkable claims.
>
> We illustrate this with two examples. In the bottom-up direction, a Markov probe with adjustable transition entropy tests whether higher training-data entropy produces more diverse generation. If the predicted trend does not appear in a matched real-model comparison, the transfer claim is rejected. Otherwise, it is accepted. In the top-down direction, degraded low-frequency token performance motivates a mixture generator with a controllable head-tail mass ratio, where each simplification step is documented so that transfer failure can be traced to a specific reduction. Probes that survive transfer validation can accumulate into a reusable catalog of instruments for testing data characteristics under consistent rules.
>
> **W2 (Example reproduces common decoding effects).**
> The example is not intended to produce a new empirical finding about decoding. Its role is to demonstrate a diagnostic capability. Probe outputs are evaluated against a known generator, which gives regime assignments with statistical meaning under the same source process. Without this calibration, a researcher observing "more diverse outputs" has no principled way to quantify or interpret how far those outputs are from the generating distribution, or to compare that deviation across models or conditions.
>
> Many diagnostic questions in Section 5 require this capability. For example, testing whether a training-data intervention shifts generation into a different typical-set regime, or whether two models trained on different data distributions occupy similar regimes, needs a ground-truth-anchored diagnostic. Standard analyses usually cannot answer these questions reliably because the ground-truth generation structure is unknown. The Markov experiment illustrates this mechanism in a minimal setting, and the four criteria operationalize extension to richer probes.
>
> Beyond standard benchmarks, probes can expose signals such as distance from generator-defined mass concentration regions, depth-conditioned failure profiles, frequency-band calibration gaps, and trajectory-level error propagation under controlled perturbations.
>
> **Q1 (How to define and evaluate probes).**
> Data probes are defined based on the four criteria outlined in our response to W1.
> The probe evaluation has two layers. The first layer is internal validity in probe space, where interventions and diagnostics must align with the target factor. The second layer is external relevance, where the same factor is tested for directional consistency in realistic LLM settings. A probe that passes only the first layer supports local probe-specific conclusions. A probe that passes both layers supports practical conclusions.
>
> **Q2 (Relationship to synthetic tasks and formal language benchmarks).**
> Synthetic tasks and formal language benchmarks are important neighboring tools. Their primary objective is usually capability characterization or comparative evaluation on fixed tasks. Data probes focus on attribution to data factors through intervention and transfer testing. In practice, many existing synthetic tasks can be embedded into the data-probe framework once control variables and diagnostics are made explicit.
>
> **Core position.** We would like to reiterate that we advocate for data probes as a **methodology** that provides a structured basis for reproducible data-centric science for LLMs. This methodology specifies how to formulate probe definitions, how to evaluate probe effectiveness, and how to calibrate claims based on transfer evidence. This directly addresses the concern that bridge claims require operational grounding. The four criteria, the two-layer evaluation, and the pre-declared failure conditions are precisely that grounding.
>
> We will include all the above clarifications and discussions in the next version of our position paper.

---

> > ### Author Rebuttal · Reviewer_rLEz · 2026-04-01
> >
> > I think some of my concerns have been addressed, but there are still some more details on this manuscript are not easily addressed in a short rebuttal, e.g., the formal definition with notations are still lacking.  Thus, I choose this option (c).

---

### Official Review · Reviewer_oJnD · 2026-03-04

**Significance:** 3
**Argument Clarity:** 4
**Rating:** 5
**Confidence:** 4

**Questions:**

1. Table 1: crux of experiment is unclear, partially because NLL's are listed for the synthetic probe case (left) but not for the model trained on natural language data (right); the reviewer assumes this is intentional, right? Therefore, one wonders how the RHS examples were actually selected; was any quantitative measure involved? It would help if a proxy for the typical set construction were defined for the real data case as well, but not sure if this is even possible. Perhaps one could consider using a different oracle LLM to assign likelihoods across a large set of samples?

To elaborate a bit further, the example data probe and experiment presented is insufficient to demonstrate a real insight that could be gained using this methodology; as a result, this choice of example unfortunately calls into question the entire premise of the probe methodology. The reviewer's appraisal is that the experiment appears to be a demonstration that varying temperature causes a model to emit sequences that deviate from the argmax/mode of the model distribution; or rather, to deviate from the typical set(s). This is almost a tautology based on the definition of how a next token prediction language model parametrizes the distribution over sequences using the chain rule of probability decomposition, and how temperature sampling is then performed over said distribution to modulate diversity during sequence generation.

2. S5 states that "Dependency Structure" could be one possible probe domain. Could the authors should try and provide a concrete example of this (and the vocab one for good measure)? The existence of probes models that mirror these more non-trivial characteristics of real data generating processes appears to be the presupposed crux of this position paper and might have been a better choice for the core example experiment.

3. The position argues for a bottom up approach, i.e. identify a real data dependence problem in a model and then from scratch design a minimal synthetic probe that provides leverage on the real problem. Perhaps a useful alternate strategy that is still roughly in scope for this position is instead to go the other way. Start by reducing examples from a real world data and behavior dependence problem that is, on its face, relatively intractable, down to a more tractable data probe model, and then study said probe. By defining the process as a reduction --- a series of coarse graining and factor elimination steps done while preserving essentials --- the path between the real world and the probe would perhaps be maintained a little more clearly. The paper reads as if identifying these linkages is a given, but this reviewer believes that it is not; indeed, this is the most non-trivial part.

4. Overall S5 touches on many good points, however, it lacks grounding to the literature. The reviewer recognizes some of the stated problems from existing studies. However the S2 "present status" vs. S5 "problems probes will solve" with no citations in the latter implicitly presents those topics as if they have not been studied previously. It would improve the position if the S5 proposed problems were connected to examples in the literature. Perhaps in each instance, a previous paper is cited with a statement about how the experimental design fits into the framework of data probes, and/or how the lack of a synth probe based framework inherently limited what the prior study could derive from its experiments. One example that comes to mind for "Scaling laws" or "Transformer Analysis with Controlled Input.... capacity" would be "How much do language models memorize?", Morris et al. 2025. They use random byte sequences with tractable entropy to build capacity scaling laws, and then identify a phase shift that reproduces for runs using natural language data. If the authors can identify existing studies that related to each one of these points, then the section and paper's impact as a whole will be greatly improved.

**Alternative Views Section:**

Yes

**Compliance With Llm Reviewing Policy A Conservative:**

Affirmed.

**Discussion Potential:**

4

**Final Justification:**

I believe that the authors' additional efforts towards formalization and contextualization in the literature as discussed will improve the presentation and reception of the work in important ways. Assuming that the additions exhibited in the anonymous PDF will appear in the camera ready draft (and the added notation developed for reviewer rLEz, though perhaps appendix for that), I'm happy to bump my score more strongly in the direction of acceptance as a quantitative indicator for the AC.

**Paper Summary:**

This paper argues that we need to develop a set of techniques based on synthetic data generation called "Data Probes" for the specific purpose of probing how ML model behaviors depend on their training data. They survey related work on our current theoretical and empirical understanding of LLMs and then discuss specific questions and research subdomains that stand to benefit from the adoption of data probes. They present an empirical case study of experiments developing and using a simple instantiation of a data probe, and then discuss a range of more realistic, pressing usecases where data probes could be brought top bear to provide real insights. They conclude with a discussion of counterpoints to their position and a call to action.

**Position:**

Yes

**Position In Title:**

Yes

**Related Work:**

4

**Strengths And Weaknesses:**

The paper discusses a pressing need in the LLM research community to develop improved experimental design tools for the data centric study of LLMs and other generative models. While they invest some effort in designing a real, implementable version of a data probe and present the results of those experiments, some weaknesses in the presentation and structure of the position's argument, and the specific (optional) choice to include a relatively toy example in the paper, shed light on why the proposed solution (data probes) may not actually end up yielding novel an significant insights in real world scenarios.

Strengths:
1. The abstract attribution problem --- the scientific study of how training data causally impacts LLM behavior --- is important, and real breakthroughs would have large implications for both research and practice.
2. The discussion of existing research (S2) and presentation of arguments for why we need to solve this abstract problem (S3) are clear and comprehensive. The section on more specific instances of problems where data probes could be leveraged (S5) is also strong, perhaps the most potentially impactful part of the paper.
3. The authors instantiate and experiment with a concrete example of their methodology (though it is toy, see related comments)

Weaknesses:
1. The argument for why "data probes" are a practical research solution to the abstract attribution problem in LLMs relies on strong presuppositions about whether the theory<->practice, toy<->industrial spectrums can be smoothly bridged by the data probe methodology. If it turns out to be impossible to construct decisive probes for real frontier problems in how specific data characteristics influence model behavior, then the position reduces to an aspirational "wouldn't it be great if we could X" even if there don't exist any roads to non-vacuous solutions for X. The reviewer argues this is one of the hardest problems in generative modeling today, and it is not clear that the position understands the largess of the presuppositions that are being made. Whether there exist probes such that laboratory results will generalize to real settings is the question at hand and the impact of the paper depends on whether or not this is the case.
2. The presented experiment somewhat underscores this primary concern; it is a toy example and the actual generalized result the experiment is meant to produce is left unclear. Some questions are included below, but in summary, the experiment describes how NLLs are calculated only in the probe case on the markov seq data, and then a pairwise qualitative comparison is shown between exemplars of each entropy class and samples from a real model. Without any quantitative measurement for the real samples that can be heuristically tied to the toy model, how are these example pairs identified? It appears to be a manually curated set of outputs, and unfortunately, this seems to exemplify the potential chasm between a synthetic probe, and some real world model behavior one might want to generalize to.
3. Despite the general relevance of all the points made in S5, this section lacks grounding to the literature. As a result, it suggests a blind spot in the position about how a large amount of existing work _does_ actually try and study some of the problems that this position suggests require the existence of data probes. The reviewer feels that the implication is that some points made here are more novel than they actually are; a simple tying in of appropriate citations would strengthen the potential significance and impact of progress on this research direction rather than detract from the work.


Current recommendation:

I think that this position paper should probably be accepted conditional on some modifications and improvements. My key concern is that the generalized form of the research problems implicated by the position are effectively "all of data attribution for LLMs". This is of course a worthwhile goal, and for this reason the paper is worthy of community discussion, but the position does not really articulate a precise (novel?) way in which theory and practice gap will be bridged using the probe methodology. Thus, it is hard to determine how much the paper adds to the conversation around how to solve real problems in data attribution. Rather, it paints a rosy picture about how "wouldn't it be cool if we could use synthetic data generators and models trained on them to explain the relation between real data and models?" and then provides a relatively toy example that doesn't clearly demonstrate bridging of this gap.

All that said, discussion potential is rated as high both because of strength 1, and also because of the debate that might surround weakness 1.

**Support:**

3

---

> ### Author Rebuttal · Authors · 2026-03-30
>
> Thanks for the constructive feedback and acknowledging the strong discussion potential.
>
> **W1 (Bridging theory and practice).** The bridge claim is stated as falsifiable outcome, not as an assumed guarantee. The core logic is problem driven reduction: 1) Start from a real data dependence question in an LLM workflow. 2) Construct a probe family that preserves the target mechanism while removing unrelated confounders. 3) Perform controlled interventions on that mechanism. 4) Test whether the direction and ordering of effects are preserved in a realistic setting. If the effects fail to transfer, the probe design is rejected for that question. This makes the bridge operational and testable instead of aspirational.
>
> **W2 (Toy experiment).** The current experiment serves as a minimal instrument check. It verifies that known-distribution diagnostics can separate regimes of generation behavior under controlled decoding changes. The intended message is not that temperature affects diversity, which is known. It is that a probe can provide calibrated regime labels tied to a ground-truth generator, then use those labels to reason about behavior patterns that also appear in practical text generation. The novelty is diagnostic identifiability under controlled data factors, not rediscovery of common decoding trends.
>
> **W3 (Section 5 grounding).** Section 5 syntheses data-centric problems. It is not as a claim that those topics are unstudied. The positioning claim is that many existing studies can be reinterpreted as partial instances of probe methodology, while a unified framework improves comparability across studies and clarifies what is missing in each setting. The practical gap is inconsistent probe definitions, inconsistent diagnostics, and weak transfer criteria across adjacent lines of work.
>
> **Q1 (Table 1 clarity).** NLL values are reported only for the probe-side outputs by design, as the true generating distribution is defined and computable only in that setting. The right-column GPT-2 outputs are based on a *randomly selected* sample for each temperature, which serve as qualitative illustrations of analogous behavioral patterns and are not presented as quantitative transfer claims. Table 1 is intended to demonstrate the feasibility to observe some *qualitatively* transferable behavior. We will revise the caption and discussion to make this clear.
>
> For quantitative transfer claims, quantitative real-text proxy could be defined using model based scores such as self perplexity, repetition entropy, etc. This proxy does not create exact typical set guarantees, but it enables controlled comparison of regime trends between probe outputs and natural text outputs. The exact implementation of such a proxy is beyond the scope of the position paper but should be studied in future work.
>
> **Q2 (Dependency and vocabulary probe examples).** A dependency probe can use a latent stack process where valid token transitions depend on nested state depth. The control knob is depth distribution and reset probability. The diagnostics include failure rate by depth, memory decay curves, and mismatch between generated and true depth-conditioned transition statistics. A vocabulary probe can use a mixture generator with head and tail token bands where head vs. tail mass ratio is controlled explicitly. The diagnostics include tail recall under fixed entropy, calibration gap by token frequency band, and robustness under tail upweighting.
>
> **Q3 (Top down reduction).** The top down strategy is compatible with the data-probe framework and can be expressed as reduction first and probing second. In particular, we propose the following top-down protocol:
> 1. Reduce a real-world problem to a tractable probe while documenting the simplification steps.
> 2. Explicitly declare which statistical properties are preserved during reduction.
> 3. Identify matching metrics applicable to real-model outputs.
> 4. Design a data probe family based on the desired statistical properties and matching metrics.
> 5. Pre-register directional predictions
> 6. State explicit falsification conditions.
>
> This approach strengthens traceability from practical problem to probe design and makes transfer failure analysis easier because each removed factor is documented.
>
> **Q4.** See answer to W3.
>
> **Core position.** Our contribution is a **methodology** that encourages reproducible data-centric science for LLMs. This methodological layer is itself a meaningful contribution for the community because it standardizes how data-centric claims are tested and compared. In the current landscape, many synthetic findings are informative but difficult to compare across papers and difficult to translate into practical conclusions. A shared probe protocol makes positive and negative transfer outcomes reportable under the same rules and turns isolated case studies into cumulative evidence that can be validated, falsified, and reused.
>
> We will add all these additional discussions to the updated paper.

---

> > ### Author Rebuttal · Reviewer_oJnD · 2026-04-01
> >
> > (AC brought to my attention that my comment was originally posted with incorrect visibility, sorry about that. Repeated here.)
> >
> > These clarifications are helpful! That said, I'd appreciate a little bit of additional detail on how the modifications will be incorporated into the updated copy before I am willing to raise my score. I think anonymous screenshots of key re-rendered sections are allowed here (check with AC if unsure). In some cases, this wouldn't matter so much, but as much of a position paper's value is derived from _how_ the position is presented versus just what the position is, it's a little more important to see how the authors intend to actually implement the points discussed across these rebuttal responses in a camera ready draft.
> >
> > **W1) and Q3)** Across the different rebuttal responses, including the one to me, you've presented a more formalized procedure and desiderata for defining and implementing data probes than was present in the original draft. This is good! please incorporate it prominently, perhaps stylized a bit using either latex or even a diagram. Relatedly, in retrospect, I don't know how distinct of a top down vs bottom up variant of the probe design process there is. However, "documenting each removed factor" and "Explicitly declare which statistical properties are preserved during reduction" to more clearly denote how the gaps between the minimal model and the real phenomenon are actually defined, feels important. I believe the "documenting" step is not really present in the bullet list given to other reviewers, nor is a concrete example of how to catalog these differences given in the current draft.
> >
> > **W2) and Q1)** The experimental setup, and table captions etc. will all greatly benefit from following the newly formalized bullets about defining "testable and falsifiable claim with a pre-declared failure condition" and the "real-world validity connection". As you can tell, that last part was not very clear to me the way the original results were presented.
> >
> > **Q2)** I actually like the PCFG examples presented to reviewer **Vp5L** better, so I'd try and include some of this instead of the more abstract example given to me.
> >
> > **W3) and Q4)** Obviously I'm biased, but I do feel strongly that Section 5 should be linked out into the literature. I actually think that this would add significant value to a broader readership if done right. The rebuttal statement "The positioning claim is that many existing studies can be reinterpreted as partial instances of probe methodology" is key, and while the reviewer and the authors seem to be aware of these connections, the person for whom this position paper will actually have the _most_ value is the researcher that does not quite realize how all these existing bits of work are related, less controlled and formalized versions of a data probe style experiment. Making these connections explicit by describing exactly how study X could be redone under the probe design methodology to make its observational findings more statistically grounded or causal is not capitalized on as fully as this reviewer would prefer.

---

### Official Review · Reviewer_vJjB · 2026-03-11

**Significance:** 3
**Argument Clarity:** 3
**Rating:** 4
**Confidence:** 4

**Questions:**

NA

**Alternative Views Section:**

Yes

**Compliance With Llm Reviewing Policy A Conservative:**

Affirmed.

**Discussion Potential:**

3

**Final Justification:**

I appreciate the author's thoughtful feedback and careful reply.

I maintain my original rating which already towards acceptance.

**Paper Summary:**

This paper argues for developing data probes. Data probes are synthetic token sequences generated from fully specified stochastic processes with known and controllable distributions, to study how data properties influence large language model behavior.

The central position is that current LLM data research relies too heavily on expensive empirical heuristics over large real-world corpora, which makes it difficult to isolate which data characteristics actually drive model performance, robustness, generalization, or failure modes. In contrast, data probes would provide a controlled and theoretically grounded experimental interface between simplified analysis and practical LLM workflows.

**Position:**

Yes

**Position In Title:**

Yes

**Related Work:**

3

**Strengths And Weaknesses:**

**Pro.**

- The paper is likely to stimulate useful discussion across theory, empirical LLM research, and interpretability.

- I think the paper is well aligned with the goals of a position track: it makes a concrete methodological proposal.

- The paper focuses on an important gap in current LLM research: while models and training algorithms are studied extensively, the role of data is still often investigated in a largely heuristic and resource-intensive way. A call for more controlled experimentation on the data side is well motivated and meaningful.

- The paper consistently argues that synthetic sequences drawn from known stochastic processes could serve as a bridge between theory and practice. The motivation is easy to follow, and the proposed benefits: control, reproducibility, exact likelihood evaluation, and lower experimental cost, are articulated clearly.

**Con.**

1. The notion of “data probe” remains somewhat broad and under-specified. The paper would be stronger if it provided a sharper definition or a set of desiderata for what qualifies as a good data probe. For example:
- Must a probe be tied to a known distribution with tractable likelihood?
- Must it support a theoretical interpretation?
- Must it exhibit transfer to nontrivial LLM behaviors?

2. Positioning against adjacent lines of work could be sharper. The paper cites several relevant areas, but the relationship to neighboring paradigms—such as synthetic algorithmic tasks, diagnostic benchmark construction, controlled distribution-shift evaluations, or other forms of artificial data used in interpretability/theory—could be clarified more explicitly (for this paper to be a really useful guideline in this area).


Overall, my recommendation for this submission is somewhere between borderline accept and accept (maybe weak accept, if possible). I like this paper, but I feel giving it too strong an ``accept" rating.

**Support:**

3

---

> ### Author Rebuttal · Authors · 2026-03-30
>
> We thank the reviewer for the thorough feedback on our position paper and for acknowledging that our paper can stimulate useful discussion.
>
> **W1 (Data probe notion).** We will incorporate a more specific definition in our updated paper: *A data probe is a family of synthetic datasets generated by a fully specified process and used to test a specific claim about model behavior under controlled data changes, where the claim can be checked and can fail.* The required criteria of a data probe include:
>
> - a known generative process,
> - controllable and interpretable knobs,
> - a measurable main metric, and
> - a testable and falsifiable claim with a pre-declared failure condition.
>
>
> Desirable criteria include diagnostics that can be computed reliably, reproducibility across seeds and model sizes, and same-direction movement on matching real-model metrics.
>
> On the question of whether a probe must transfer to nontrivial LLM behavior, we treat probe validity and real-world relevance as separate properties. Validity requires controlled sensitivity and specificity on the probe setting. Relevance requires at least one pre-registered real-model metric with movement in the same direction as the probe-side prediction. For claims about real LLM behavior, if no pre-registered real-model metric follows the predicted direction, we label the result synthetic-only and do not claim transfer. If outcomes are mixed across seeds and stability rules are not met, we report the result as inconclusive. A simple practical check is as follows: If we cannot name the knob, main metric, direction, and failure condition in one sentence, it is not a probe claim. Exact likelihood is a strong diagnostic option but not the only valid one, and a synthetic benchmark that changes multiple factors at once without isolating one knob-to-endpoint claim falls outside the scope of a data probe.
>
> **W2 (Positioning relative to adjacent paradigms).** We agree that a clearer taxonomy would strengthen the paper. Synthetic algorithmic tasks are mainly about whether the model can solve a designed task. Diagnostic benchmark construction focuses on surfacing where performance fails on designed test sets. Controlled distribution-shift evaluation studies whether performance holds when test data changes. Data probes, in contrast, link a data property to a behavior change under controlled conditions, which supports clearer attribution.
>
> For example, a distribution-shift test can show a performance drop but often cannot identify which part of the shift caused it, whereas data probes are designed to vary one property at a time. A benchmark that varies prompt length and label noise simultaneously between samples is not a data probe because attribution to a claimable factor is not possible. Although we acknowledge a possible alternative view that careful benchmark design alone may be sufficient, but our position is that shared reporting rules are still needed for cross-work comparison and falsification.
>
> A unified data-probe methodology reduces fragmentation across different works that use incompatible synthetic setups and incompatible metrics. This improves reproducibility because probe definitions and intervention knobs are precise. It also improves interpretability because diagnostics are tied to known generator factors, as well as communication across theory and empirical communities by giving both sides a shared object for experimentation and analysis. We will make the taxonomy and differences clear in the revised paper.

---

> > ### Author Rebuttal · Reviewer_vJjB · 2026-04-02
> >
> > I appreciate the author's thoughtful feedback and careful reply.
> >
> > I maintain my original rating which already towards acceptance.

---

### Official Review · Reviewer_Vp5L · 2026-03-12

**Significance:** 3
**Argument Clarity:** 3
**Rating:** 5
**Confidence:** 5

**Questions:**

1. How do data probes concretely differ from existing synthetic-data approaches for studying transformers (e.g., [1], [2]) and in-context learning (e.g., [3])?

2. What generative processes beyond first-order Markov chains do you envision for capturing hierarchical and compositional structure in language while remaining theoretically tractable?

3. How would data probes apply to RL-based post-training (e.g., RLHF, GRPO) and reasoning, where the relevant data involves preferences or chain-of-thought rather than sequences from a fixed distribution?

[1] Makkuva, A. V., Bondaschi, M., Girish, A., Nagle, A., Jaggi, M., Kim, H., and Gastpar, M. Attention with Markov: A curious case of single-layer transformers. ICLR, 2025.

[2] Rajaraman, N., Bondaschi, M., Makkuva, A. V., Ramchandran, K., and Gastpar, M. Transformers on Markov data: Constant depth suffices. NeurIPS, 2024.

[3] Shivam Garg, Dimitris Tsipras, Percy S Liang, and Gregory Valiant. What can transformers learn in-context? a case study of simple function classes. NeurIPS, 2022.

**Alternative Views Section:**

Yes

**Compliance With Llm Reviewing Policy A Conservative:**

Affirmed.

**Discussion Potential:**

3

**Final Justification:**

I am raising my score to 5, as I trust that the camera-ready version will include comparisons with other LLM studies that conduct scientific experiments using synthetic data, as well as a discussion on how to handle post-training methods such as RLHF. That said, I strongly sympathize with the other reviewers' concerns regarding how much the synthetic-to-real gap can be reduced.

**Paper Summary:**

This position paper advocates for developing systematic methodologies for generating synthetic sequences from known random processes, termed "data probes," to study how data characteristics affect LLM behavior. The authors argue that because data probes come from fully specified distributions, researchers can precisely control and measure statistical properties (entropy and etc), enabling reproducible and resource efficient experiments. They connect this framework to information-theoretic concepts like typical sets and provide a preliminary experiment using a GPT-2 small model trained on Markov chain-generated sequences, demonstrating that the typical set framework can characterize LLM generation behavior (typical set and etc) across different decoding strategies. The paper discusses potential applications spanning data analysis and mechanistic interpretability.

**Position:**

Yes

**Position In Title:**

Yes

**Related Work:**

2

**Strengths And Weaknesses:**

**Strengths**:

1. Well motivated problem. Understanding how data properties fundamentally affect LLM behavior is an important open problem. The paper correctly identifies that current approaches rely heavily on expensive trial-and-error with large datasets and lack principled methodology.

2. Clear and well organized presentation. The paper systematically builds its argument. The writing quality is generally high.

3. Broad applicability. Section 5 presents a convincing range of potential applications (data diversity, overfitting and etc), suggesting the framework could serve as a versatile research tool.

**Weaknesses**

1. Unclear differentiation from existing work. Several cited works (Makkuva et al., 2025; Rajaraman et al., 2024) already use Markov chain-based synthetic data to study transformers. The paper acknowledges these but does not clearly articulate what the "data probes" framework adds beyond giving a unifying name to what multiple research groups are already doing. The paper would benefit from a dedicated comparison explaining how data probes differ in scope, methodology, or goals from these existing approaches, as well as from the in-context learning literature that uses synthetic function classes (e.g., [1], [2], [3]) to study transformers in controlled settings.

2. Scalability concerns. The current demonstration uses a 1st-order Markov chain with 128 states and GPT-2 small. The gap to practical LLMs (billions of parameters, vocabularies of 50K+) is substantial. More critically, natural language exhibits hierarchical and compositional structure that Markov chains fundamentally cannot capture. The paper does not adequately address whether data probes can scale to capture these properties, or what classes of random processes would be needed.

3. Limited coverage of the LLM workflow. The paper claims data probes can be useful across "training, tuning, alignment, in-context learning, etc." But does not discuss how they apply to RL-based post-training (e.g., RLHF, GRPO) or reasoning processes, which are critical stages of modern LLM development. It is unclear how synthetic sequences from known distributions could probe preference learning, reward modeling, or reasoning capabilities.


[1] Shivam Garg, Dimitris Tsipras, Percy S Liang, and Gregory Valiant. What can transformers learn in-context? a case study of simple function classes. NeurIPS, 2022.

[2] Satwik Bhattamishra, Arkil Patel, Phil Blunsom, and Varun Kanade. Understanding in-context learning in transformers and llms by learning to learn discrete functions. ICLR, 2024.

[3] Jaeyeon Kim, Sehyun Kwon, Joo Young Choi, Jongho Park, Jaewoong Cho, Jason D. Lee, Ernest K. Ryu. Task Diversity Shortens the In-Context Learning Plateau. TMLR, 2025.

**Support:**

2

---

> ### Author Rebuttal · Authors · 2026-03-30
>
> We thank the reviewer for the thorough feedback on our position paper.
>
> **W1 & Q1 (Differentiation from existing work).** We do not position “data probes” as a new name for existing settings. Our position is that we need a **shared experimental protocol** for data-centric analysis to understand LLMs, where synthetic generators are selected and evaluated using unified criteria instead of ad hoc choices per paper. Prior works such as Markov-transformer analyses and in-context-learning function classes provide important ingredients. However, the difference is that data probes are not a specific synthetic family. It is a **reusable data methodology**.
>
> Existing works typically optimize one axis, such as theoretical tractability, in-context learning characterization, or benchmark behavior. In contrast, the data-probe methodology uses those ingredients inside a unified pipeline that starts from a data hypothesis, introduces controlled interventions, measures diagnostics tied to those interventions, and checks transfer trends on practical LLM behaviors.
>
> Specifically, a setup counts as a data-probe methodology only if all these four elements are present:
> 1. There is an explicit generator with at least one known control variable.
> 2. It has intervention knobs aligned with the hypothesis under test.
> 3. It has measurable diagnostics linked to those knobs, with exact likelihood when available and principled proxy otherwise.
> 4. It has transfer-oriented validation to connect theory and practice.
>
> Moreover, an important added requirement over prior work is pre-declared failure rules for transfer claims. If a practical observation does not align with what the theory predicts, that outcome must be reported as evidence against the transfer claim. This also distinguishes our work from a survey article. We propose a protocol with explicit pass/fail structure rather than only summary of prior results. We will add these discussions along with a dedicated comparison in the related work section.
>
>
> **W2 & Q2 (Scalability).** The first-order Markov experiment is a low-complexity instantiation of the data-probe pipeline for a preliminary illustration. It is not the end state of a fully developed data-probe methodology. We provide the preliminary result in this position paper to facilitate further discussions.
>
> The same data-probe protocol scales to richer generators with stronger linguistic structure. For example, we could use *probabilistic context-free grammar (PCFG) probes* to capture hierarchy and compositional depth, *higher-order / latent-state sequence processes* to capture longer memory, *programmatic compositional generators* to capture rule composition and algorithmic structure, *mixture/curriculum generators* to capture domain overlap and controlled distribution shift, etc. This addresses the practical gap to modern LLM training by retaining intervention control while increasing structural fidelity.
>
> A tractable progression is as follows: 1) Markov, 2) higher-order Markov/HMM, 3) PCFG/latent grammar, 4) compositional program generators. Each step increases expressiveness while preserving at least one analyzable handle, such as closed-form likelihood, approximate posterior structure, or exact generator semantics. For example, PCFG probes expose explicit tree depth and branching factors, latent-state probes expose controllable memory length and mutual information, and program generators expose compositional operators with exact ground-truth execution traces. This provides a principled complexity ladder instead of a binary toy-vs-real divide.
>
> **W3 & Q3 (RL-based post-training and reasoning).** The probe logic extends naturally to post-training scenarios. For RLHF/reward modeling, *preference probes* can be sampled from latent utility models with controlled annotator noise, subgroup bias, and preference inconsistency. For example, a latent utility process can generate pairwise rankings so that reward-model identifiability and bias sensitivity can be evaluated directly. For GRPO/RL post-training, *policy probes* with known reward landscapes and rollout dynamics isolate optimization pathologies (reward hacking, instability, exploration collapse) under controlled settings. For reasoning, *trajectory probes* with known latent dependency graphs make it possible to localize failure stages (state tracking, compositional step execution, error propagation) instead of only measuring final-answer accuracy. Such multi-step supervision signals can be used to test whether improvements come from better decomposition, better memory over steps, or better calibration under branching uncertainty. This extends probes from "token distribution analysis" to end-to-end post-training diagnostics.
>
> We will include these additional discussions in the next version of the paper.

---

> > ### Author Rebuttal · Reviewer_Vp5L · 2026-04-03
> >
> > I thank the authors for the detailed rebuttal. The four criteria for what constitutes a data probe help sharpen the definition and partially address my concern about differentiation from existing work. The proposed complexity ladder and extensions to post-training are also sensible directions.
> >
> > That said, I remain partially unconvinced on W1. The rebuttal frames data probes as a reusable methodology rather than a specific synthetic family, but many existing works already follow a similar structure, they define controlled generative processes, vary specific parameters, and draw falsifiable conclusions. The added value of formalizing this as a unified protocol is plausible but not yet demonstrated in the paper. A dedicated comparison showing how specific prior studies do or do not satisfy the four criteria would make this distinction concrete.

---

### Decision · Program_Chairs · 2026-04-30

**Decision:**

Accept (regular)

**Comment:**

There was a consensus towards acceptance after the rebuttal period (4/4/5/5), with reviewers praising the work for highlighting the critical need for a principled, systematic methodology to study how data characteristics causally affect LLM behavior. Although several reviewers initially raised concerned e.g. regarding the lack of a strict formal definition for "data probes", unclear differentiation from existing synthetic data literature, and the scalability of the preliminary toy Markov experiment, those were all successfully addressed by the extensive rebuttal from the authors. Even if the practical impact of such synthetic data probes remains to be validated in the real world at scale, I believe it is clearly interesting and relevant enough to the ICML community to warrant acceptance at the conference.